# Response of Wheat, Maize, and Rice to Changes in Temperature, Precipitation, CO_2_ Concentration, and Uncertainty Based on Crop Simulation Approaches

**DOI:** 10.3390/plants12142709

**Published:** 2023-07-20

**Authors:** Mengting Qin, Ennan Zheng, Dingmu Hou, Xuanchen Meng, Fanxiang Meng, Yu Gao, Peng Chen, Zhijuan Qi, Tianyu Xu

**Affiliations:** 1School of Hydraulic and Electric Power, Heilongjiang University, Harbin 150080, China; 2212032@s.hlju.edu.cn (M.Q.); 2222032@s.hlju.edu.cn (D.H.); 2222012@s.hlju.edu.cn (X.M.); gaoyu501@126.com (Y.G.); 2021007@hlju.edu.cn (T.X.); 2College of Agricultural Science and Engineering, Hohai University, Nanjing 210098, China; chenpeng_isotope@163.com; 3School of Water Conservancy and Civil Engineering, Northeast Agricultural University, Harbin 150030, China; zhijuan.qi@neau.edu.cn

**Keywords:** climate change, crop yield, adaptation measures, uncertainty

## Abstract

The influence of global climate change on agricultural productivity is an essential issue of ongoing concern. The growth and development of wheat, maize, and rice are influenced by elevated atmospheric CO_2_ concentrations, increased temperatures, and seasonal rainfall patterns. However, due to differences in research methodologies (e.g., crop models, climate models, and climate scenarios), there is uncertainty in the existing studies regarding the magnitude and direction of future climate change impacts on crop yields. In order to completely assess the possible consequences of climate change and adaptation measures on crop production and to analyze the associated uncertainties, a database of future crop yield changes was developed using 68 published studies (including 1842 samples). A local polynomial approach was used with the full dataset to investigate the response of crop yield changes to variations in maximum and minimum temperatures, mean temperature, precipitation, and CO_2_ concentrations. Then, a linear mixed-effects regression model was utilized with the limited dataset to explore the quantitative relationships between them. It was found that maximum temperature, precipitation, adaptation measure, study area, and climate model had significant effects on changes in crop yield. Crop yield will decline by 4.21% for each 1 °C rise in maximum temperature and increase by 0.43% for each 1% rise in precipitation. While higher CO_2_ concentrations and suitable management strategies could mitigate the negative effects of warming temperatures, crop yield with adaptation measures increased by 64.09% compared to crop yield without adaptation measures. Moreover, the uncertainty of simulations can be decreased by using numerous climate models. The results may be utilized to guide policy regarding the influence of climate change and to promote the creation of adaptation plans that will increase crop systems’ resilience in the future.

## 1. Introduction

A major and ongoing issue is global climate change, which has an effect on food production and human health [1]. The Sixth Assessment Report (AR6) of the United Nations Intergovernmental Panel on Climate Change (IPCC) reported that the global surface temperature (GST) for 2001–2020 is 0.99 °C higher than that for 1850–1900 and that global surface temperature increases in the 21st century will exceed 1.5 °C and 2 °C unless significant global reductions in CO_2_ and other greenhouse gas emissions are implemented in the upcoming decades [2]. Climate change will have an even greater negative influence on agricultural systems due to the frequency and intensity of extreme weather occurrences [3]. Climate has a significant influence on agricultural productivity since the factors that drive crop development include water, atmospheric temperature, and photosynthetically effective radiation [4,5].

Many crop models, such as the Decision Support System for Agrotechnology Transfer (DSSAT), Agricultural Production Systems Simulator (APSIM), and Environmental Policy Integrated Climate (EPIC), have been extensively utilized to examine how climate change may affect agricultural production [6,7,8,9]. Compared to traditional field experiments, model simulations have the advantages of being less expensive and more efficient and involving easier to control variables. They can model the processes of crop development and yield generation in response to alterations in climatic circumstances, soil properties, and management techniques. Arunrat, et al. [10] used the EPIC model to examine how climate change would affect the water footprint and production of major crops in northern Thailand. Houma, et al. [11] utilized the AquaCrop model to investigate the influences of climate, irrigation practices, and weed control measures on rice yields in the Malaysian lowlands. Chisanga, et al. [12] utilized the APSIM-Maize and CERES-Maize models in combination with five climate models and two emission scenarios to simulate how climate change may affect future maize yields in Zambia.

However, inconsistencies in crop models in terms of the interaction of temperature, seasonal rainfall, and greenhouse gases may lead to different or even opposite trends in predicted crop production in the context of future climate change [13,14]. In addition, uncertainties may come from climate models, study areas, and emission scenarios and whether CO_2_ fertilizer and adaptation measures are considered [3,15]. Jiang, et al. [15] integrated several crop models (CMs), species distribution models (SDMs), and global climate models (GCMs) for the local harvest of winter wheat on the Loess Plateau of China and found that CMs predicted winter wheat yields with more certainty than GCMs and SDMs when future changes in crop climate suitability were taken into consideration. Xiong, et al. [16] measured the contribution of models, climate, parameterization, and management to the total uncertainty in the forecasted production response to warming, concluding that the overall uncertainty was much greater at middle and high latitudes than at low latitudes and that crop models generated more uncertainty than the other sources combined. In summary, a thorough analysis of the effects of future climate change on agricultural yields is obviously required since these uncertainties have resulted in contradictory projections of the response of crop yields to climatic variations.

Systematic reviews of the effects of climate change on crop growth began in the 1980s, and the extensive research generated on this basis has been condensed in numerous scientific reviews and IPCC reports [17]. Knox, et al. [18] provided a systematic overview of the projected impact of climatic factors on the yields of seven important crop categories grown in Europe. Liu, et al. [17] systematically evaluated the influences of climatic factors on crop production in China and emphasized significant variations in consequences between particular crops and geographic regions, suggesting that wheat yields respond most significantly to temperature increases, followed by rice and maize. Meta-analysis is a method of quantitative investigation that offers an organized and thorough study of the literature on a certain issue and permits a summary of the results of previous research [19]. Previous multiple regression models for the individual level focused on inter-individual variation and individual characteristics, ignoring inter-group variation, and therefore ignoring group-level characteristics at the individual level, whereas the use of mixed-effects models allows the systematic analysis of sub-individual and group-level effects, examining how macro-variables modulate the effects of micro-variables, and whether group-level explanatory variables affect the effects of individual-level explanatory variables. Although there is a substantial body of research on the effects of climate change on agricultural yields, there are relatively few comprehensive studies, which poses an obstacle to the assessment of climate change impacts.

Considering the influence of climate change on crop yields at a global scale, we built a database of 1842 samples based on a systematic review of 68 studies published globally since 2000. This study addresses the following issues: (1) the impact of varying degrees of variation in climatic factors (temperature, precipitation, and CO_2_ concentration) and adaptation measures on the yield of major crops; (2) the degree and trends of changes in crop yields over the future period; and (3) sources of uncertainty in climate change impact assessments. The findings of the study help shed light on the variations in crop production forecasts among studies and serve as a guide for further research in the field.

## 2. Materials and Methods

### 2.1. Literature Selection and Data Collection

To conduct a comprehensive evaluation of how climate change is affecting agricultural production (wheat, maize, and rice), a detailed search was conducted of all relevant studies that have researched the influence of climate change on crop growth. The literature search was conducted utilizing the keywords “climate change”, “effect”, “crop/wheat/maize/rice”, “yield”, “production”, “growth”, and “model” to search and filter articles published after 2000 in the China National Knowledge Infrastructure (CNKI), Web of Science, Science Direct, and Google Scholar databases about how future climate change will affect agricultural productivity. The following were the criteria used to choose the literature: (1) no studies evaluating the influence of historical climate change on crop production were included; instead, only the literature evaluating the influence of future climate change on crop production was included; (2) studies had to be based on quantitative assessments of crop models, excluding studies that used field trials or agricultural test stations where only one climatic factor was manipulated to determine the degree of production impact; (3) only the literature that predicted crop yield changes was included, and this literature included studies that predicted directional changes and concrete percentage change values; and (4) studies that assessed the impact of at least one meteorological factor, including temperature (maximum, minimum, and average temperatures), precipitation, and CO_2_ concentration changes, were included.

The literature’s title served as the basis for the initial screening; the abstract served as the basis for the secondary screening, and only the complete texts of the articles that met the screening requirements were then examined. We finally obtained 68 relevant references.

The chosen research typically described variations in crop yields predicted by crop models under various climate conditions in one or more locations. The climate scenarios studied revealed potential future climate conditions given the matching historical or present climate conditions (baseline scenarios). Notably, each article’s baseline period was slightly different. Examples include 1980–2010 for Osman, et al. [20] and 1982–2012 for Srivastava, et al. [21]. There were no significant differences in the baseline period of the articles included in our study. For every combination of article/study area/crop model/climate model and scenario, the relative percentage change in average yield (RCY) was computed with Formula (1):(1)$$\mathrm{RCY}=100\times \frac{\left(\mathrm{Future}\;\mathrm{average}\;\mathrm{yield}-\mathrm{Baseline}\;\mathrm{average}\;\mathrm{yield}\right)}{\mathrm{Baseline}\;\mathrm{average}\;\mathrm{yield}}$$
where “Future average yield” and “Baseline average yield” represent crop yield values averaged across years for both future and baseline climate situations, respectively.

The relative percentage change in crop yields for various study areas, various climate scenarios, and various predicted years in every study then served as a sample of observations (for instance, Zheng, et al. [6] estimated the percentage change in rice yields for one study site under three future climate scenarios, SSP126, SSP245, and SSP585, for the 2040s, 2060s, and 2080s, which provided 3 × 3 × 1 = 9 sample observations).

Based on the sample processing method described above, data were pooled from 68 selected papers, resulting in a final sample size of 1842 (full dataset), including 777 for wheat, 587 for maize, and 478 for rice. The difference between crop yield change and zero change in each country was compared utilizing the Student’s *t* test. The samples were then divided based on meteorological factors (changes in temperature, precipitation, and CO_2_ concentration), the presence of adaptation measures, the combination of climate patterns (single or complex), and the time period, to identify factors that influence the response of crop yields to climate change. Simultaneously, the changes in meteorological factors linked to every sample observation were taken from the research as follows: change in maximum and minimum temperature (°C), change in precipitation (%), and change in CO_2_ concentration (ppm) compared to baseline and whether adaptive strategies were taken into consideration. The full combining sample of observations for these variables in some publications was 447 (constrained dataset), as at least one of these factors was absent. We performed a descriptive analysis of the complete dataset and used the constrained dataset to build meta-regression models for quantitative analysis to quantify the relationship between climate variables and crop yields.

### 2.2. Descriptive Analysis

The complete dataset (i.e., all available data for every climate variable) was used to plot graphs to investigate the relationship between percentage change in crop yield (RCY) and variations in temperature (maximum temperature, minimum temperature, and mean temperature) (°C), changes in mean precipitation (%), and changes in CO_2_ concentration (ppm). To examine the response of the RCY to these five climate variables, we fitted the data using a local polynomial with a 95% confidence interval.

### 2.3. Quantitative Analysis

The influences of temperature, precipitation, CO_2_ concentration, and adaptation measures on crop yield variability were estimated using constrained datasets. The majority of the initial studies in our work predicted multiple values of crop yield change simultaneously, leading to the nonindependence of crop yield projections in the initial studies and failing to evaluate the study hypotheses properly. In addition, because different studies provided different sample sizes, this led to heterogeneity. To address these issues, a linear mixed-effects model was utilized to estimate the equations, with fixed effects including the maximum temperature change $\left(\Delta T_{max}\right)$, the minimum temperature change $\left(\Delta T_{min}\right)$, average precipitation change $\left(\Delta P\right)$, CO_2_ concentration change $\left(\Delta CO_{2}\right),$ and categorical factors indicating adaptation measures (A: yes or no) and random effects including study areas, crops, crop models, climate models, and scenarios. The model was calculated based on Formula (2):(2)$$\mathrm{RCY}=\alpha_{1}\times \Delta T_{max}+\alpha_{2}\times \Delta T_{min}+\alpha_{3}\times \Delta P+\alpha_{4}\times \Delta CO_{2}+\alpha_{5}\times A+\beta_{0}+\beta_{study\;areas}\;+\beta_{crops}+\beta_{crop\;models}+\beta_{climate\;models}+\beta_{climate\;scenarios}+\varepsilon$$
where the RCY is the relative percentage change in crop yield (%); $\Delta T_{max}$ is the maximum temperature change (°C); $\Delta T_{min}$ is the minimum temperature change (°C); ΔP is the mean precipitation change (%); $\Delta CO_{2}$ is the CO_2_ concentration change (ppm); A is the adaptation measure (A indicates two categorical factors that equal 1 when the sample takes into account the effect of the adaptation measure; otherwise, it equals 0); $\alpha_{1},\;\;\alpha_{2},\;\;\alpha_{3},\;\;\alpha_{4},\;and\;\alpha_{5}$ are the four regression coefficients in the fixed-effects section; $\beta_{0}$ is the intercept; $\beta_{study\;areas},\;\beta_{crops},\;\;\beta_{crop\;models},\;\;\beta_{climate\;models},\;\beta_{climate\;scenarios}$ are the random effects of the study areas, crops, crop models, climate models, and climate scenarios, respectively; and ε is the residual random term. The framework of the study is depicted in Figure 1.

## 3. Results

### 3.1. Distribution of Datasets

The distribution of the study areas covered by the complete dataset is given in Figure 2. The complete dataset of the 68 studies collected reported the relative percentage change in crop yield (RCY) for each study region under different future climatic scenarios. The average crop yield change (%) reported for each country is summarized in Figure 3, showing data regarding all locations, crop modeling, climate models, climate scenarios, and time periods. The graph shows that the data collected are distributed over 20 countries, including China, Australia, India, and Pakistan.

Table 1 summarizes the minimum, mean, and maximum crop yield percentage changes recorded in the collected research for each country and applies the Student’s *t* test to determine the significance of the RCY relative to zero change. The RCY ranged from −12.90% (wheat in Pakistan) to 47.67% (maize in South Africa). Overall, the average wheat yields increased by 6.66%, the average maize yields increased by 10.02% and the average rice yields decreased by 1.49%. With the exception of Australia, there were significant differences in relative wheat yield changes compared to zero change in other countries. Additionally, there were significant differences in relative yield changes compared to zero change for maize in India, Pakistan, Ethiopia, United States, and China and relative yield changes compared to zero change for rice in Haiti, Malaysia, and China.

### 3.2. Relationship between Climatic Factors and Crop Yield Changes

The relationship between the relative percentage change in crop yield (RCY) and variations in maximum temperature (°C), minimum temperature (°C), average temperature (°C), and precipitation (%) is shown in Figure 4. The graphs show that the response of the main crop (wheat, maize, and rice) relative yield changes to variations in temperature and precipitation was significantly different. The response of relative yield changes to changes in maximum and minimum temperatures for the same crop showed a more consistent trend. When the maximum temperature rose by more than 2 °C and the minimum temperature rose by more than 1.2 °C, there was a significant positive impact on wheat yields. For maize, when warming was below 3.5 °C, the change in maize yield showed a decreasing trend, but it was still greater than zero. When the maximum temperature was higher than 3.5 °C and the minimum temperature was higher than 3.9 °C, the maize yield showed an increasing trend. However, for rice, changes that were too high or too low in terms of the maximum and minimum temperatures reduced rice yield, and the optimal temperature was approximately 2.5 °C (Figure 4a–f). Wheat yield tended to increase when the average temperature change was greater than 0.9 °C. A certain degree of average temperature increase increased maize and rice yields, while excessive average temperature change led to a trend of reduction in maize and rice yields (Figure 4g–i). Temperature is the main meteorological element influencing crop yield, and the crop growth process requires optimum temperatures. Asseng, et al. [22] suggested that, under climate change conditions, water shortages caused by high temperatures may be a major cause of crop yield reduction.

In addition, the response of RCY (%) to changes in precipitation (%) cannot be ignored. Overall, an increase in precipitation of more than 0.2% had a positive effect on wheat yield. In contrast, changes in average precipitation (%) showed contrasting trends on changes in maize and rice yields (Figure 4j–l).

The relationship between CO_2_ concentration and RCY is shown in Figure 5. It can be seen that a certain degree of increase in CO_2_ concentration had a positive influence on crop yields, in some scenarios, compensating for yield losses due to climatic parameters; however, in certain scenarios, the influence was not considerable, depending mainly on the degree of climatic variables (temperature and precipitation). Increased atmospheric CO_2_ concentrations may have direct effects on crops in the form of CO_2_ fertilization [23], through increasing photosynthesis, decreasing stomatal conductance, and increasing water use efficiency [24].

The values of crop yield changes for pairs with and without CO_2_ fertilization are plotted in Figure 5d. Each pair of values came from the same research location, crop model, and future climate scenario. The extent of the influence of CO_2_ concentration relied heavily on the factors taken into account (such as the research location, crop type, time period, and emission scenario). Although increased CO_2_ concentrations may increase crop yields, it is expected that increased water stress due to warmer temperatures will counteract the CO_2_ fertilization function, which may result in a general decline in agricultural production [25].

### 3.3. Impact of Adaptation Measures on Crop Yields

Pairs of data samples with and without adaptation measures were selected for this study (other climate variables were the same), and Figure 6 represents the distribution of crop yield changes for the adapted and unadapted measures. The influence of climate change on crop yields can be mitigated with some adaptation measures, with the average crop yield increasing by 14% with adaptation measures and decreasing by 6.04% without adaptation measures. The average change in crop yield with adaptation measures ranged from −11.71% to 270%. The range without adaptation measures was −63% to 42.5%.

Adaptation techniques on the basis of crop model study are often autonomous adaptations to current cropping systems, for instances, changes in sowing dates, varieties, irrigation, and fertilization strategies. The detrimental influences of climate change on agricultural production are somewhat countered by these adaptations. Ding, et al. [26] mitigated the influences of climate on rice growth in China through appropriate irrigation and sowing date adjustments. Huang, et al. [3] adapted maize to future climate change by adjusting planting dates and changing varieties. Olabanji, et al. [27] used a combination of planting date changes and adequate irrigation to increase crop yields. Despite the immense potential of adaptation measures to improve crop yields, there is still a significant degree of ambiguity regarding the impact and efficiency of adaptation methods, with the major sources of uncertainty depending on the study site and the type of adaptation strategy employed [3]. Therefore, it is important to explore measures to address climate change that are regionally appropriate according to actual regional conditions.

### 3.4. Impact of Climate Models and Time Periods on Crop Yields

The predicted effect of future climate change on crop productivity is subject to a variety of uncertainties. Among these uncertainties are the selection of climate model and the impact of the future time period, which cannot be ignored. Figure 7 shows the change in percentage crop yields reported in the study using complex (≥2) and single (=1) climate models. The findings of this study indicate that the use of multiple climate models reduced the median range of relative crop yield changes and outliers compared to the use of a single climate model. Therefore, by using more climate models, crop estimates may be calculated with less uncertainty.

The total quantity of RCY samples in various ranges is shown in Figure 8, with different colors indicating the range of RCY variation. The results show that, in the 2020s, 2030s, 2040s, 2050s, 2060s, 2070s, 2080s, and 2090s, the number of samples with a positive RCY was 0.56, 0.67, 0.62, 0.53, 0.76, 0.44, 0.62, and 0.59 of the total number of samples, respectively. In addition, the differences in RCY (%) between time periods are also shown in Figure 8, and the results show that the average crop yield changes in the 2020s, 2030s, 2040s, 2050s, 2060s, 2070s, 2080s, and 2090s were 2.98%, 10.59%, 2.46%, 3.73%, 19.52%, 13.73%, 4.25%, and 14.18%, respectively; the median crop yield changes were 1.37%, 6.71%, 2.63%, 0.68%, 12.26%, −3.1%, 2.76%, and 7.33%, respectively. The Student’s *t* test indicated that crop yield change was significantly different from the zero change in all time periods except for the 2070s.

### 3.5. Regression Analysis of the Restricted Dataset

The restricted dataset was modeled with a linear mixed-effects model containing complete information on crop yield variation (%), temperature variation ($\Delta T_{max}$ and $\Delta T_{min}$, and °C), precipitation variation (ΔP, %), and CO_2_ concentration variation ($\Delta CO_{2}$, ppm). The dependent variable for the model was the RCY values from 447 samples, and the assessment outcomes are displayed in Table 2. The findings of the assessment showed that the effects of maximum temperature, precipitation, and adaptation measured on RCY were significant. The coefficient of $\Delta T_{max}$ was markedly negative, indicating that, for each 1 °C rise in maximum temperature, the crop yield decreased by approximately 4.21%. However, a rise in the minimum temperature had a positive influence on crop yields, probably since low temperatures can expose crops to low-temperature cold damage during the reproductive period. Increases in precipitation and CO_2_ concentration had a positive influence on increases in crop yields, with each 1% rise in precipitation and each 1 ppm rise in CO_2_ concentration increasing crop yields by 0.43% and 0.02%, respectively. The crop yields with adaptation methods were 64.09% greater than those without adaptation measures, indicating that the influence of adaptation measures was more substantial than other meteorological factors. The estimates for the random effects of study area, crop, crop model, climate model, and climate scenario were 172.68, 177.26, 10.28, 173.64, and 19.81, respectively. The study area and the climate model were statistically significant (*p* < 0.05), indicating that the variation in predicted crop yields was high between the study areas and climate models.

## 4. Discussion

### 4.1. Impacts of Future Climate Change on Crop Yields

The meta-analysis method was adopted to determine the thresholds for climate change and to study the influences of changes in temperature, precipitation, CO_2_ concentration, and adaptation measures on global crop yields. The findings of the linear mixed-effects model based on a constrained dataset suggest that increased precipitation, CO_2_ concentration, and specific adaptive management strategies can compensate for the negative influences of increased temperature on crop yields.

Climate explains 30–50% of global crop yield variation [28]. Temperature and precipitation are considered to be the main meteorological factors controlling regional and global crop growth [29]. Future climatic changes are anticipated to result in increased heat and moisture stress as well as a detrimental influence on agricultural productivity. The length of plant cycles will also be shortened due to global warming [30]. Our study demonstrates that crop productivity is significantly influenced by maximum temperature, precipitation, and the use of adaptation strategies. Crop yield will decline by 4.21% for every 1 °C rise in the maximum temperature, while it would increase by 0.43% for every 1% rise in precipitation, and crop yield with adaptation measures increased by 64.09% compared to crop yield without adaptation measures. There are two main reasons for reduced crop yields at high temperatures: higher temperatures accelerate crop phenological development and shorten the growing period [31], and high-temperature stress that occurs during the crucial reproductive stage (post-shoot stage) reduces grain quality and promotes spikelet sterility [32]. Higher temperatures alone, according to a meta-analysis, decreased rice yields by an average of 33% [33]. In many situations, appropriate management measures such as sowing dates, variety improvement, and effective water and fertilizer management techniques can help to mitigate the negative influences of climate change on crops [26,27]. Raes, et al. [30] have shown that, when optimal management conditions are assumed, yields can even be doubled or tripled in some cases. The results of the systematic statistical analysis by Aggarwal, et al. [34] showed that the actual yield changes owing to climate change were substantially larger than the predicted yield changes, suggesting that the impact of technical developments on yield increases appears to be significant compared to the adverse consequences of climate change and that, once these measures are taken, climate change may not significantly increase food production challenges in most regions, except for a few potential hotspots distributed around the world. Furthermore, in our study, a rise in minimum temperature had a relatively positive influence on crop yield, because low-temperature stress reduces leaf area and stems, limits the leaf photosynthetic system, and can seriously reduce seed output, notably in varieties that are vulnerable to cold temperatures [35].

In addition, our study of paired samples with and without the CO_2_ fertilization effect found that the CO_2_ fertilization effect had a nonnegligible role in the response of crop yields to future climate change, as evidenced by the positive response of 14.89% of crop yields to climate change without the CO_2_ fertilization effect and by the more optimistic future crop yield situation when the CO_2_ fertilization effect was considered. The positive value was 71.28%. Elevated atmospheric CO_2_ concentrations can directly alter physiological processes in crops (e.g., photosynthesis and stomatal conductance) [36]. The results of the analysis of Vanuytrecht, et al. [37] also show that increased CO_2_ concentrations have a significant influence on various macroscopic growth processes in crops and have a positive influence on water productivity and crop yield.

### 4.2. Uncertainty in the Assessment of Future Climate Change Impacts

In our study, variations in the study areas and climate models were found to have a significant impact on crop yields (*p* < 0.05). Garofalo, et al. [38] found that geographical location (continental vs. Mediterranean) had a significant impact on the ranking of each factor when analyzing and ranking the uncertainty of influencing factors. Xiong, et al. [16] examined the global geographic distribution of uncertainty in climate change effect forecasts, showing lower uncertainty at low latitudes than at middle and high latitudes. Wang, et al. [13] also highlighted site-specific sources of uncertainty in agricultural production forecasts under future climate change. There are many reasons for regional differences, the primary determinants being climate, soil types, geography, and level of agricultural management [18,39]. However, extreme outcomes from climate models with a wide range of attributes can create uncertainty and may not properly predict future climate circumstances [4,40]. Multiple studies have also shown that global climate models (GCMs) are a major source of uncertainty [41,42]. It is crucial to employ various GCMs when examining how climate change affects agricultural systems. Araya, et al. [40] utilized 2 crop models, 20 GCMs, and 2 climate scenarios to evaluate the influence of climate change on future maize yields in Ethiopia and indicated that uncertainty in terms of the influence of climate change on maize originated mainly from climate models and crop models. Differences in crop yields due to climate scenarios are mostly caused by differences in the extent of the projected climate change [4]. However, the differences between the emission scenarios were small and may have been due to the interaction of factors, such as higher temperatures, reducing the yield increase caused by CO_2_ fertilization effects. Asseng, et al. [43] also showed that simulation models, study areas, climate models, projected CO_2_ concentrations, and other factors make future estimates extremely unclear.

This study utilized a meta-analysis approach to aggregate the results of existing studies, and this approach can significantly improve the reliability and generalizability of the findings. However, we must acknowledge that there is still some uncertainty in the projections. Agricultural growth models and climate models are frequently integrated in agricultural research since it is frequently not possible to determine the effects of future climate change on crop yield through field experiments. However, while they simplify the sophistication of the actual world, they also tend to bring in significant uncertainty concerns [44,45,46]. Therefore, the scope of our study was restricted to evaluating the modeling outcomes of a wide range of climate change impact studies, each of which inherently includes a variety of effect adjustments or heterogeneity-causing factors. These studies, for instance, take into account a variety of various scales and nations, as well as various emission scenarios, downscaling techniques, ecological circumstances, and crop variety, agricultural systems, and the level of agronomic technology. Thus, it is inevitable that the simulated yields achieved merge the anticipated effects of climate change and the various combined influences of several other factors that are latent in each study. Despite these drawbacks and uncertainties, research on how climate change affects crops enables the results of all available modeling studies to be pooled in a methodical and objective way to reach broad but crucial judgments [4,18,34].

In addition, it should be noted that there are data collection restrictions. For example, predicted crop yield data are not available for all areas of the world. In addition, our data included only peer-reviewed journals prior to the search period, and new research may be published in the future. Although fresh research may not have a big impact on the results, it might further support current figures. We recommend that subsequent studies that include new modeling data be included in our dataset to keep our findings current.

Despite the limitations, direct comparisons of the influences of climate change and other factors on agricultural system are extremely challenging, as noted by Challinor, et al. [47]. Combining crop yield effect research is a first step in estimating the extent and uncertainty of climate change, and conclusions based on existing findings have some credibility but are not sufficiently accurate. More comprehensive and accurate studies of crop yield projections under changing environments are still needed in the future.

## 5. Conclusions

This study is based on a meta-regression approach, using the literature to analyze and quantify crop yields under future climate change. The findings of this study show that temperature, precipitation, CO_2_ concentration, and adaptation measures have an impact on crop yields, especially maximum temperature, precipitation, and adaptation measures. For each 1 °C rise in maximum temperature, crop yield will decrease by 4.21%; for each 1% rise in precipitation, crop yield will increase by 0.43%; and crop yield with adaptation measures increases by 64.09% compared to crop yield without adaptation measures. When the impact of CO_2_ fertilization is taken into account, the future crop yield situation will be more positive.

In addition, a comparison of the percentage change in crop yields simulated by simple and complex climate models demonstrated that the uncertainty in simulations may be decreased by using more climate models. The findings of the mixed-effects model demonstrated that variations in the study areas and climate models had a significant influence on crop yield changes. The results may be used to guide policy on the influences of climate change and to promote the creation of adaptation plans that will increase agricultural systems’ resilience in the future.

## Figures and Tables

**Figure 1 plants-12-02709-f001:**
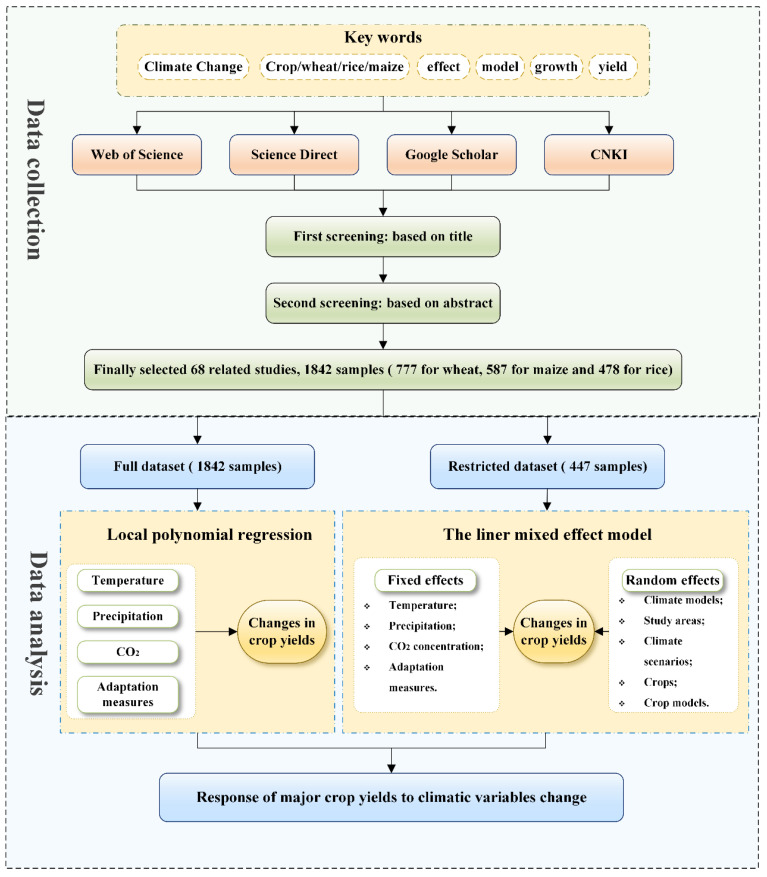
Framework of this study.

**Figure 2 plants-12-02709-f002:**
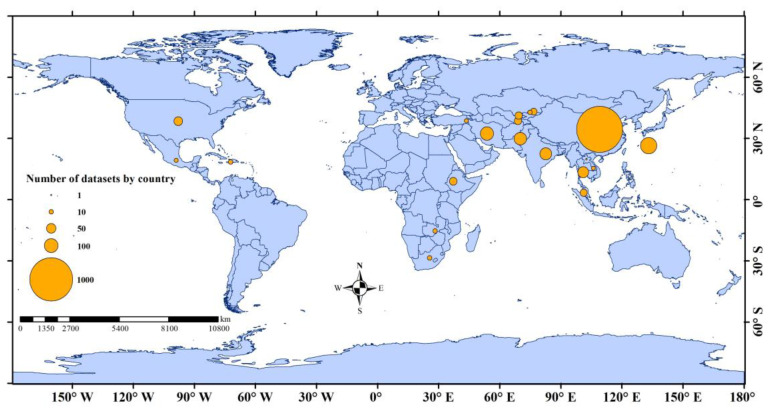
Distribution of the study areas.

**Figure 3 plants-12-02709-f003:**
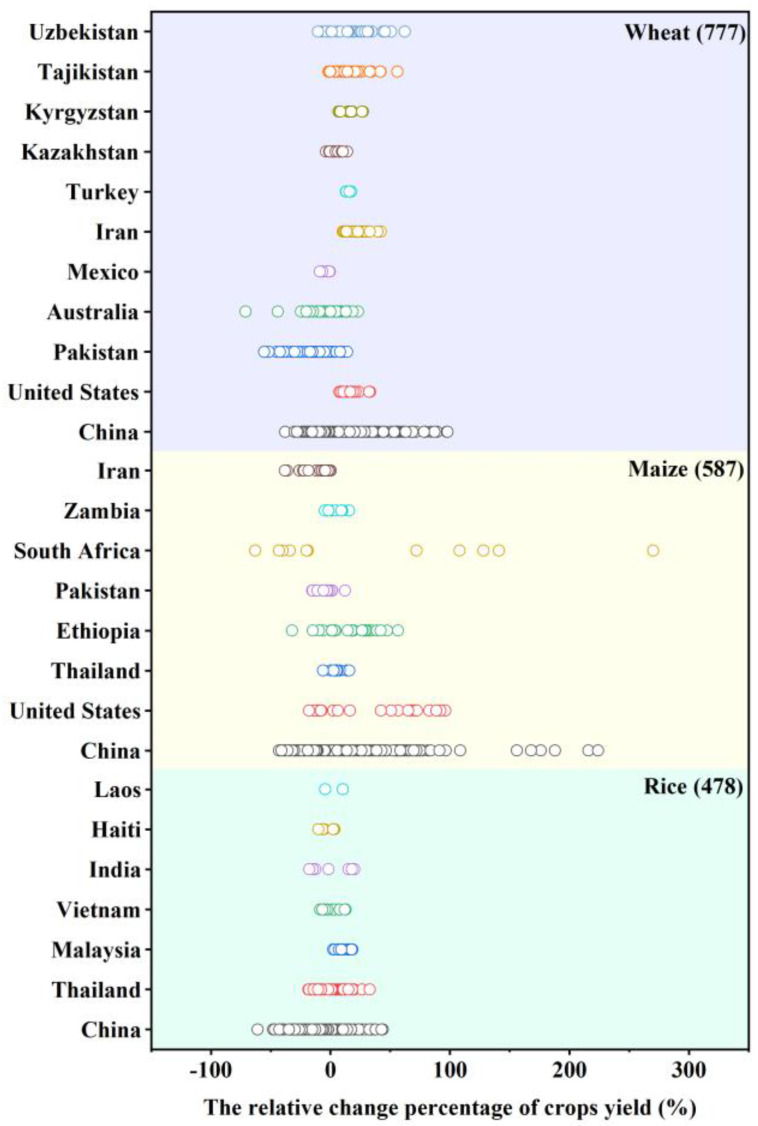
Overview of projected crop yield changes (%) in every country.

**Figure 4 plants-12-02709-f004:**
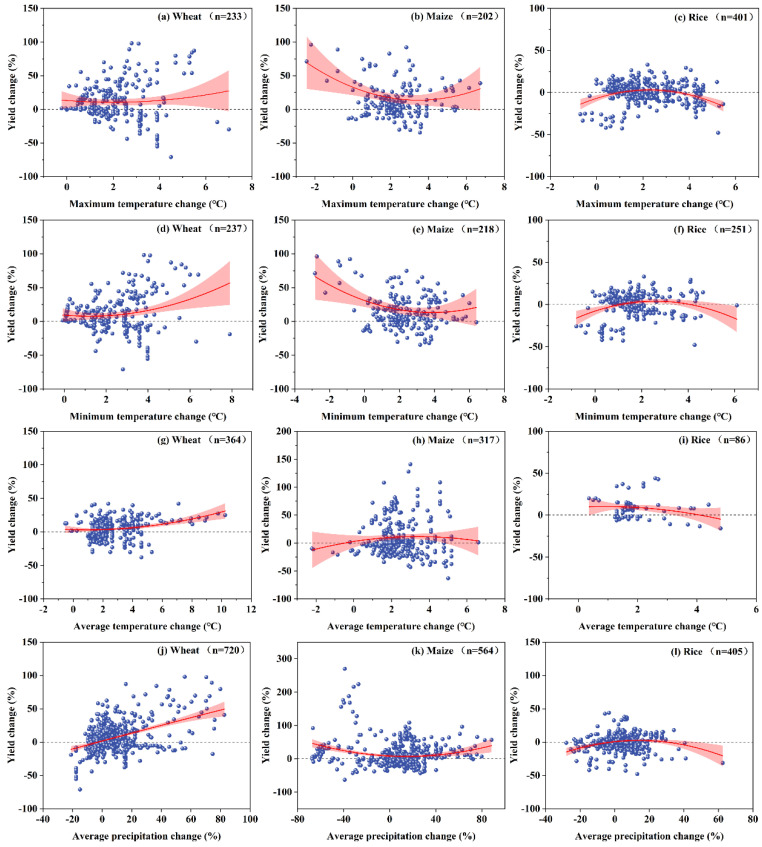
Relationships between percent change in crop (wheat, maize, and rice) yield with temperature (maximum temperature (**a**–**c**), minimum temperature (**d**–**f**), and average temperature (**g**–**i**)) and average precipitation change (**j**–**l**).

**Figure 5 plants-12-02709-f005:**
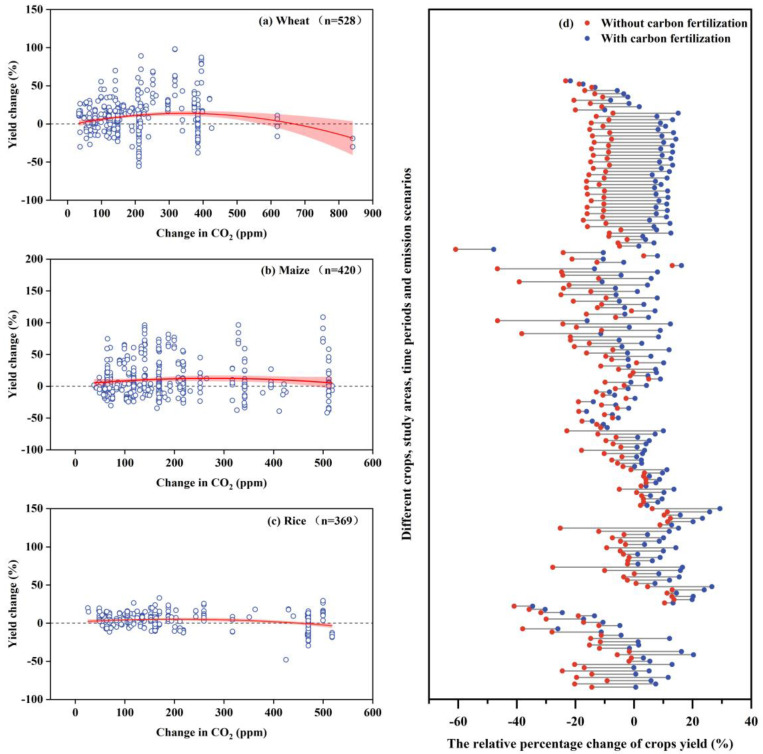
CO_2_ concentration change (ppm) (**a**–**c**) and RCY (%) in response to the presence or absence of CO_2_ fertilization (**d**).

**Figure 6 plants-12-02709-f006:**
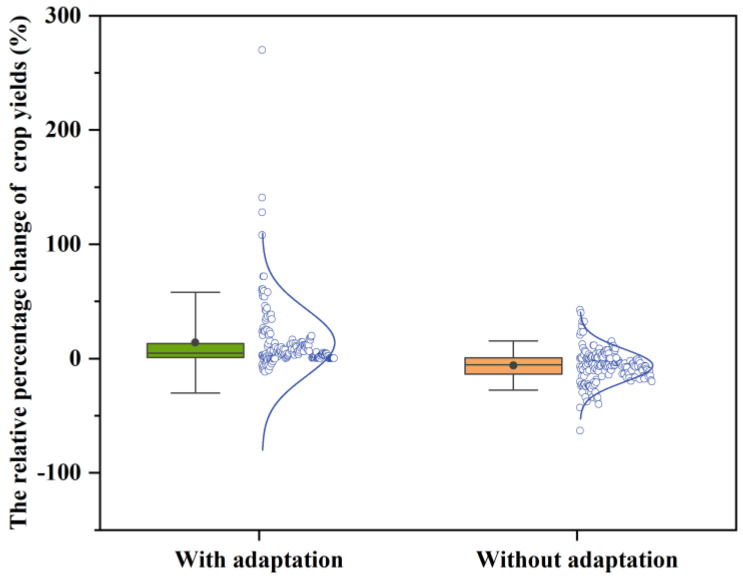
Box and normal curves of the distribution of RCY (%) with and without adaptation measures. The box boundaries indicate the 25th and 75th percentiles; black lines and dots represent the median and mean; and whiskers below and above the box indicate the mean ± 1.5 SD, respectively.

**Figure 7 plants-12-02709-f007:**
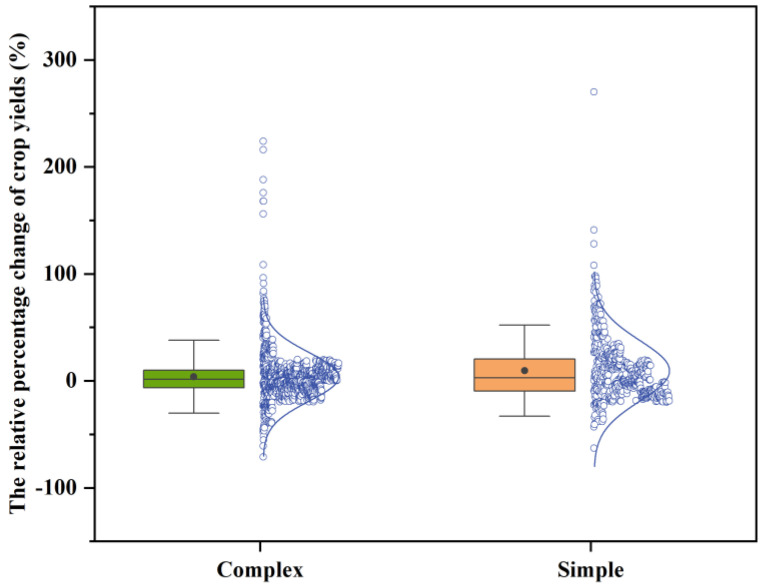
Box and normal curves of the projected changes in average yield (%) for all crops and time slices, aggregated by climate change modeling approach (complex (*n* ≥ 2), simple (*n* = 1)). The box boundaries indicate the 25th and 75th percentiles; black lines and dots represent the median and mean; and whiskers below and above the box indicate the mean ± 1.5 SD, respectively.

**Figure 8 plants-12-02709-f008:**
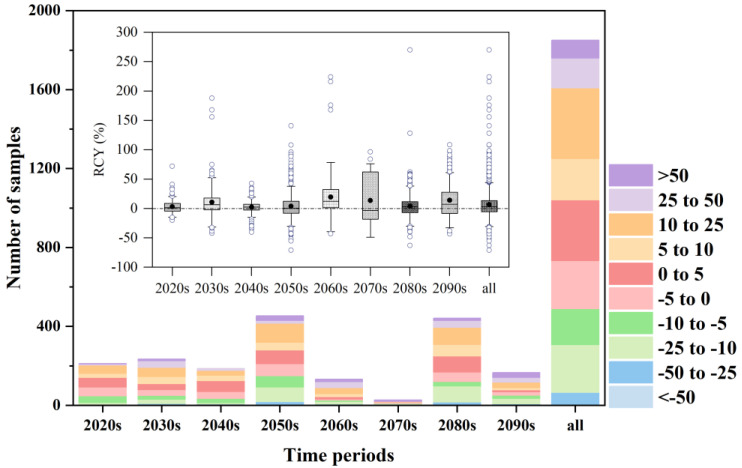
Distribution of the relative percentage change in crop yield (RCY) with time period.

**Table 1 plants-12-02709-t001:** Summary of reported RCY (%) regarding all simulation results in every country.

	Country	Minimum	Mean	Maximum	Student’s *t* Test	
					*p* Value	S/NS
Wheat	Uzbekistan	−10.37	20.06	62.28	<0.001	S
	Tajikistan	−1.54	14.59	55.93	<0.001	S
	Kyrgyzstan	6.70	15.18	27.27	<0.001	S
	Kazakhstan	−3.69	4.89	14.13	<0.001	S
	Turkey	12.94	15.16	17.25	<0.001	S
	Iran	10.50	20.32	41.91	<0.001	S
	Mexico	−8.74	−4.55	−0.50	0.008	S
	Australia	−71.00	−0.70	22.96	0.444	NS
	Pakistan	−55.41	−12.90	14.10	<0.001	S
	United States	7.60	16.38	33.10	<0.001	S
	China	−37.85	8.89	98.10	<0.001	S
	All	−55.41	6.66	98.10	<0.001	S
Maize	Iran	−38.10	−6.88	0.20	<0.001	S
	Zambia	−4.65	3.28	15.48	0.155	NS
	South Africa	−63.00	47.67	270.00	0.132	NS
	Pakistan	−15.20	−4.54	12.13	0.015	S
	Ethiopia	−32.17	13.73	56.58	<0.001	S
	Thailand	−6.30	4.10	15.79	0.053	NS
	United States	−18.00	29.58	96.07	0.002	S
	China	−43.03	10.23	224.00	<0.001	S
	All	−63.00	10.02	270.00	<0.001	S
Rice	Laos	−4.30	3.00	10.30	0.752	NS
	Haiti	−10.00	−3.42	3.35	0.032	S
	India	−17.67	1.27	20.00	0.800	NS
	Vietnam	−8.59	0.47	12.60	0.816	NS
	Malaysia	2.41	11.36	18.49	<0.001	S
	Thailand	−18.50	2.18	32.99	0.182	NS
	China	−60.91	−2.62	43.90	<0.001	S
	All	−60.91	−1.49	43.90	0.005	S

Notes: S = significant; NS = not significant.

**Table 2 plants-12-02709-t002:** Results of the linear mixed-effects model.

Fixed Effect				Random Effect			
Variable	Coefficient	t Value	*p*	Variable	Estimate	z Value	*p*
$\Delta T_{max}$	−4.21	−2.48	0.013 *	Study areas	172.68	2.70	0.007 **
$\Delta T_{min}$	1.03	0.57	0.571	Crops	177.26	0.97	0.332
ΔP	0.43	6.71	<0.001 ***	Crop models	10.28	0.34	0.731
$\Delta CO_{2}$	0.02	1.54	0.125	Climate models	173.64	2.00	0.046 *
A (no = 0; yes = 1)	64.09	4.78	<0.001 ***	Climate scenarios	19.81	1.59	0.112
Intercept	69.73	4.75	<0.001 ***	Residual	191.27	13.76	

Notes: significance levels: *p* < 0.05 *, *p* < 0.01 **, *p* < 0.001 ***.

## Data Availability

The data for this study are available from the corresponding author upon reasonable request.

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
