# Peer review of "Response of Wheat, Maize, and Rice to Changes in Temperature, Precipitation, CO2 Concentration, and Uncertainty Based on Crop Simulation Approaches"

_plants, 2023, doi:10.3390/plants12142709_

Round 1

Reviewer 1 Report

The paper titled “Response of main crop yields to future climate change and uncertainty based on crop simulation approaches” based on a meta-regression approach, using literature to analyse and quantify crop yields under future climate change. The findings of the study show that temperature, precipitation, CO2 concentration, and adaptation measures have an impact on crop yields, especially maximum temperature, precipitation, and adaptation measures.

I think that this publication can be called not only "Review" but these are original methods and studies using only literature data.

The authors undertook the enormous challenge of finding the literature they were interested in and thoroughly researching it. They took into account 68 published studies and conducted a thorough analysis of them.

The presented research deserves appreciation. It is of course easier to make a model for one location (country) and one species. The authors analyzed studies from multiple locations and considered 3 main crop species.

Title

The title is well formed. You can consider whether, however, instead of "main crop" not insert specific plant species. This will avoid repeating the word "crop" twice in the title.

I'm also wondering about the word "future" - but isn't climate change already happening today? Might drop the word "future"

 Abstract

Line 12 - I consider the allegation that there is a lack of precision in the studies considered to be misplaced.

 Introduction

There is a well-chosen literature - consistent with the content of the work.

Material and methods.

Well described

Results

Correctly

 Discussion

Correctly

Concclusion

Correctly

Reference

Well chosen for the research topic and uses the latest literature.

Author Response

We thank the reviewers for the recognition given to this manuscript and for their valuable comments. The main corrections in the paper and the responds to the reviewer's comments are as following:

Point 1: The title is well formed. You can consider whether, however, instead of "main crop" not insert specific plant species. This will avoid repeating the word "crop" twice in the title.

I'm also wondering about the word "future" - but isn't climate change already happening today? Might drop the word "future".

Response 1: Thanks for your comments. We have made the following changes in the title: first, replace "main crops" with "wheat, maize, and rice", and second, remove the word “future”.

Point 2: Line 12 - I consider the allegation that there is a lack of precision in the studies considered to be misplaced.

Response 2: Thank you for your comments. We deleted " Due to the complexity of the dynamic environment, the reported research lack precision. ".

Reviewer 2 Report

Manuscript entitled „Response of main crop yields to future climate change and un-2 certainty based on crop simulation approaches” submitted to Plants journal is well written and the results are presented in a logical and coherent manner.

The paper is adequately organized and the topic is interesting and focuses on a meta-regression approach, using literature to analyse and quantify crop yields under future climate change, so it is important for the success of cultivation in an unstable climate. Therefore, I believe that the results of the research review carried out deserve to be published in a widely available format.

Although the manuscript is well-edited, however small improvements should be introduced that will improve its quality:

- The numerical data in the tables are given with too much accuracy. With a three-digit value (table 1 and 2 and abstract) three decimal places are unnecessary.

- An explanation of the abbreviation RCY should be included in the caption of Figure 8

- The abstract is partly a repetition of the conclusions. It should be relevant to the entire article.

Author Response

We thank the reviewers for the recognition given to this manuscript and for their valuable comments. The main corrections in the paper and the responds to the reviewer's comments are as following:

Point 1: The numerical data in the tables are given with too much accuracy. With a three-digit value (table 1 and 2 and abstract) three decimal places are unnecessary.

Response 1: Thank you for your comments. We reduced three decimals to two decimals (table 1 and 2 and abstract).

Point 2: An explanation of the abbreviation RCY should be included in the caption of Figure 8.

Response 2: Thank you for your comments. We included an explanation of RCY in the caption of Figure 8.

Point 3: The abstract is partly a repetition of the conclusions. It should be relevant to the entire article.

Response 3: Thank you for your comments. We have made some changes to the abstract to better relevant it to the entire article.